

# Using the tracer flux ratio method with flight measurements to estimate dairy farm CH4 emissions in central California

Conner Daube[1], Stephen Conley[2,3], Ian C. Faloona[3], Claudia Arndt[4], Tara I. Yacovitch[1], Joseph R. Roscioli[1], Scott C. Herndon[1]

[1]Aerodyne Research, Inc., Billerica, MA, 01821, USA
[2]Scientific Aviation, Inc., Boulder, CO, 80301, USA
[3]Department of Land, Air, and Water Resources, University of California, Davis, CA, 95616, USA
[4]Environmental Defense Fund, San Francisco, CA, 94105, USA

*Correspondence to*: Conner Daube (daube@aerodyne.com)

**Abstract.** Tracer flux ratio methodology was applied to airborne measurements to quantify methane ($CH_4$) emissions from two dairy farms in central California during the summer. An aircraft flew around the perimeter of each farm measuring downwind enhancements of $CH_4$ and a tracer species released from the ground at a known rate. Estimates of $CH_4$ emission rates from this analysis were determined for whole sites and sub-sources (animal housing, liquid manure lagoons). Whole-site $CH_4$ flux rates for each farm, Dairy 1 (5,850 ± 793 kg $CH_4$ day$^{-1}$, 95% confidence interval) and Dairy 2 (3,699 ± 685 kg $CH_4$ day$^{-1}$, 95% confidence interval), closely resembled findings by established methods: ground-based tracer flux ratio and mass balance. Sub-source emission rates indicate a greater fraction of the whole-site emissions come from liquid manure management than animal housing activity, similar to bottom-up estimates. Despite differences in altitude, we observed that the tracer release method gave consistent results when using ground or air platforms. With no consideration of this analysis methodology during the experimental design, two groups were able to perform three methods quantifying $CH_4$ emitted from these dairy farms over a few summer days.

## 1 Introduction

Methane ($CH_4$) released into the atmosphere as a result of agricultural activity, such as enteric fermentation and anaerobic digestion, significantly contributes to overall greenhouse gas emissions in the United States (USEPA, 2017). The California Air Resources Board (CARB) attributes approximately 60% of recent $CH_4$ emissions in California to agriculture, with 45% of $CH_4$ emissions directly related to dairy farm activity for 2013 (CARB, 2017). Reduction strategies proposed by CARB seek to reduce California's $CH_4$ emissions to 40% below 2013 rates by 2030 (CARB, 2017), thereby emphasizing the need for accurate methods to directly quantify the contribution of different $CH_4$ sources within agricultural operations. Estimates of $CH_4$ emissions due to dairy livestock can be calculated using inventory emission factors combined with activity data on animal populations, animal types, and details about feed intake in a particular country (IPCC, 2006). Other methods to estimate $CH_4$ emissions from ruminants involve direct atmospheric measurements. Emissions from dairy farms have been estimated in the Los Angeles basin, California, using downwind airborne flux measurements (Peischl et al., 2013). Farm-scale measurements of $CH_4$ have been made using a variety of techniques and instruments, such as open-path infrared spectrometers (Leytem et al., 2017), tunable-



infrared direct absorption spectroscopy (Hacker et al., 2016), and column measurements employing solar absorption spectrometers with comparisons to cavity ring-down spectrometers (Viatte et al., 2016). Several studies of various $CH_4$ sources (e.g. natural gas pipelines, landfills, dairy farms) assert that inventory-based calculations tend to underestimate emissions compared to atmospheric observations and modelling (Brandt et al., 2014; Miller et al., 2013; Peischl et al., 2013; Trousdell et al., 2016).

Atmospheric studies have often used specific gases as tracers to distinguish a sample of interest from background conditions or interferences. Tracer gases released at known rates have been employed in experiments looking at chemical transport (Ferber et al., 1986), dispersion (Record and Cramer, 1958), source allocation (Lamb et al., 1995; Mønster et al., 2014), and model verification (Sykes et al., 1983) using mobile laboratories (Wang et al., 2009; Yacovitch et al., 2015), radiosondes, sampling towers, and ground-based equipment. Application of tracer gases in agricultural studies have involved insertion of a sulfur hexafluoride ($SF_6$) permeation tube into the rumen of a cow with subsequent collection of time-integrated breath samples (Grainer et al., 2007). Inverse-dispersion techniques have employed line-source releases of $SF_6$ within a dairy farm combined with open-path measurements to understand whole-site emissions (McGinn et al., 2006). Release of a tracer gas directly into the atmosphere, 2-3 m above ground level, can be used to determine and distinguish $CH_4$ emissions from various sources within a site (Roscioli et al., 2015). This study quantifies $CH_4$ emissions using the well-established tracer flux ratio method at two dairy farms over the course of 8 summer days (Lamb et al., 1995; Roscioli et al., 2015). Controlled releases of tracer gas from various areas on each farm mixed with site-derived emissions were observed by an instrumented aircraft and mobile laboratory (Arndt et al., 2018). Using this technique provided the flexibility to estimate entire dairy farm emissions, apportion emissions among sub-sources (animal housing, liquid manure management, etc.), and to determine methane:carbon dioxide ($CH_4$:$CO_2$) ratios on multiple scales.

Uncertainty in measurements from low-flying airborne studies has been attributed to the need to extrapolate results below the minimum safe flight heights (~150 m) as regulated by the Federal Aviation Administration (Conley et al., 2017; Hacker et al., 2016). Prior to this study, Aerodyne Research, Inc. (ARI) performed controlled ground-releases of ethane ($C_2H_6$) in Colorado and Arkansas, while Scientific Aviation (SA) made measurements above in a similar aircraft to the one used in this study (Conley et al., 2017). The original release rate of $C_2H_6$ was estimated via a refined mass balance technique, with a +2% difference observed during tests in Colorado (50 laps flown) and +24% difference in Arkansas (19 laps flown) as described in Conley et al. (2017). These releases did not correspond to any $CH_4$ source (natural gas site, dairy farm, etc.), but demonstrated the feasibility of using a low-flying aircraft to successfully quantify flow rates from controlled tracer gas releases. Using tracer flux ratio in this study, we again utilized the aircraft to detect emitted tracer gas and then compared with dairy farm emissions to evaluate $CH_4$ emission rates.

The field study was originally focused on estimating $CH_4$ emissions from dairy farms and distinguishing on-site sources using established techniques (Arndt et al., 2018). An intentional effort was made to align measurement time windows of the mobile laboratory and aircraft for the purpose of inter-comparison between the tracer flux ratio and mass balance methods. As a result, the aircraft was exposed to several hours of ground-released tracer gas. Due to this overlap in time, we were able to (1) further assess the viability of observing enhanced concentrations of a ground-



released tracer gas from an aircraft at low flow rates, (2) compare $CH_4$ and $C_2H_6$ enhancements emitted from dairy farms via tracer flux ratio, and (3) directly compare the application of tracer flux ratio methodology to simultaneous ground and airborne measurements.

## 2 Project description

### 2.1 Participants

In a collaborative effort, SA and ARI attempted a flight-based tracer release experiment to quantify $CH_4$ emissions from two dairy farms in central California. This study reanalyzes data collected as part of an Environmental Defense Fund project that occurred in June 2016 (Arndt et al., 2018). Both groups performed established techniques in the field to estimate dairy farm emissions. ARI employed tracer flux ratio methodology with two tracer gases and a mobile
laboratory, while SA conducted a mass balance experiment from a light aircraft.

Aerodyne Research, Inc. (ARI) drove ground-based transects in a mobile laboratory (miniature Aerodyne Mobile Laboratory, "minAML") equipped with highly precise Aerodyne Tunable Infrared Laser Direct Absorption Spectrometers (TILDAS) measuring a variety of species ($CH_4$, $C_2H_2$, $C_2H_6$, and CO). A Li-Cor (Lincoln, Nebraska, USA) non-dispersive infrared (NDIR) instrument (Model 6262) measured $CO_2$ and $H_2O$. Meteorological and
positional data (wind, temperature, relative humidity, barometric pressure, and GPS) were collected at all tracer release sites and on the vehicle, using multiple AIRMAR 200WX WeatherStation® (Milford, NH, USA) instruments and a Hemisphere V103 GPS Compass (Scottsdale, AZ, USA). To minimize drift and maintain accurate baseline values on the TILDAS instruments in the minAML, a valve sequence enabled overblowing of the inlet with ultra-zero air every 15 minutes for 45 s (including cell purging). A full description of the equipment used during this project can be found
in the Supplementary Information of Arndt et al. (2018).

The initial study prioritized overlapping time periods of data collection between all participants for the sake of inter-comparison. Doing so ensured that ARI would be releasing tracer gas during SA flights. Since SA also had a TILDAS on-board measuring $C_2H_6$ during these times, it was possible to treat these flights as a tracer release experiment similar to that performed with the ground-based equipment. During each flight, SA recorded multiple
species ($CH_4$, $C_2H_6$, $O_3$, $CO_2$, and $H_2O$), atmospheric conditions (horizontal winds, temperature, and RH), and positioning information (altitude, latitude, and longitude).

During this study, the aircraft flew low and close to the sites, which had a combination of spread-out point source emitters (cows) and large open area sources (anaerobic lagoon and settling cells). SA conducted 11 flights over 6 days, usually flying twice a day, in the late morning and mid-afternoon. Flights typically lasted 1-2 h for a given farm, flying
in spirals looping around the perimeter of the farm housing and manure management areas. ARI measured for 3 days at Dairy 1 and 5 days at Dairy 2. The mobile lab drove at several different times of day for each site, trying to capture any diurnal effect, but always overlapping with the aircraft at least once a day.



## 2.2 Tracer release

Tracer gases, ethane and acetylene, were released from ground-based tripods (2-3 m high) at a variety of locations on the dairy farms with the intention of co-locating with known emission sources (animal housing, anaerobic lagoons, settling cells, etc.). Tracers were used in an effort to distinguish and to quantify sources, by positioning them within

each respective emission area. Often, each tracer was released at a single point from each major sub-source, typically the liquid manure management (anaerobic lagoon and settling cells) and animal housing areas (barns and lots). For this study, only the position and release rate of ethane is relevant (Fig. 1). Release rates of $C_2H_6$ ranged from 10–40 slpm throughout the project (averaged 15 slpm). Detailed descriptions of the tracer flux ratio technique used during this work can be found in Arndt et al. (2018) or more generally in Rosicoli et al. (2015). In summary, tracer gas

released close to a source produces a plume that experiences the local wind dynamics and meteorological conditions akin to the nearby emission of interest, thereby proving a representation of those emissions. A plume is considered to be a co-located enhancement above ambient concentrations of $CH_4$ and tracer gas. Active tracer release overlapped with on-site flight transects for approximately 11 h during this week-long project. Exact timing of the overlap between the release of $C_2H_6$ and sampling periods by the aircraft is shown in Table 1.

Ethane was selected over other gases due to the lack of potential interference with nearby sources and long atmospheric lifetime. At one of the two sites, $C_2H_6$ from a small well pad (~2.5 km from closest point of farm) could be observed on the ground at close distances. This interference was characterized and eliminated using its measured $C_2H_6$:$CH_4$ ratio (Yacovitch et al., 2014) in combination with wind direction and farm layout.

## 2.3 Data quality assurance

Analysis of tracer flux data involves comparing slopes or areas of enhancements between tracer gas and site $CH_4$ emissions. Linear regression of the time-aligned $CH_4$ and $C_2H_6$ results in a molar ratio ($CH_4$:$C_2H_6$). The molar ratio scaled by the amount of tracer gas released, determines a $CH_4$ emission rate for the specific plume encounter. Due to the speed of the aircraft (typically ~65 m s$^{-1}$), observations of plume emissions were brief. On average, identified plumes lasted 12 seconds (8 s for Dairy 1; 15 s for Dairy 2), not including a significant amount of time collected before

and after enhancements to ensure accuracy of baseline calculations during analysis. On a plume-by-plume basis, the integrated area method proved inconsistent for many cases, so linear regression was used exclusively (Roscioli et al., 2015).

Prior to analysis, all data had appropriate calibration factors applied, correcting minor deviations in flow control by mass flow controllers and instrument performance for specific species. Instrument calibrations occurred in the field

at several times during this campaign using mixed-gas standards diluted with ultra-zero air. Distance between tracer release locations and aircraft position was determined using the basic trigonometry. Uncertainties for emission rate estimates are determined as 95% confidence intervals.

Plumes observed by the aircraft were included in the analysis after meeting certain criteria. Requirements included: tracer gas flowing on site for more than 10 minutes prior to observation, correlated plumes of $CH_4$ and $C_2H_6$

based on high coefficient of determination from a least-squares fit ($R^2 > 0.5$), and positive enhancements above baseline for $CH_4$ and $C_2H_6$. After meeting these standards, each plume was viewed and additional conditions were





manually considered: wind direction and speed (as recorded on the aircraft and on-site), duration of the enhancement, validity of the linear regression fits, location of the aircraft relative to the sources, and correlation between $CH_4$ and other species ($CO_2$, CO, $C_2H_6$) indicating interferences or source allocation.

## 3   Results

### 3.1   Flight conditions by site

While flying transects around each site, plumes of $CH_4$ and $C_2H_6$ were observed as frequently as once per minute. Short-lived enhancements ranged hundreds of ppb for $CH_4$ (typically ~200-300 ppb) and sub-ppb for $C_2H_6$ (typically ~0.5-1.5 ppb). Figure 2 depicts an example plume event during a transect at Dairy 1 with correlated enhancements of $CH_4$ and $C_2H_6$ as the aircraft passed to the SW of the site.

At each dairy farm, the plane gradually flew a sequence of stacked circles around the facility with an average radius of ~900 m depending on the ratio of the strength of the horizontal wind to the surface heating (Conley et al., 2017.) At Dairy 1, flights went as low as 79 m above ground level (AGL), while achieving a maximum altitude of 1,244 m AGL. Fly-overs at Dairy 2 went even lower, with minima between 33 and 56 m AGL, and consistently reached heights of ~550 m AGL. Flying at low altitudes improved the signal-to-noise ratio for $C_2H_6$, helping to partially compensate for the relatively low release rates. Wind direction varied at Dairy 1 between the morning (NW) and afternoon (SW), generally building in strength throughout the day (~3 – 4.5 m s$^{-1}$) as is common in the Central Valley due to the diurnal thermal forcing of the vast mountain-valley circulation (Zhong et al., 2004). Dairy 2, situated farther into the San Joaquin Valley, experienced consistent NNW winds that were sampled on days with a slightly greater average speed (~6 m s$^{-1}$).

Dairy 2 consisted of a long rectangular area of animal housing, made up of large free stall barns and open lots. In the northeast of the farm, an open-air manure lagoon was set just north of two long settling cells. Larger than Dairy 2, Dairy 1 had more free stall barns and open lots. Separated from the animal housing, a large lagoon and settling cell extend side-by-side to the north of the barns. Detailed descriptions of meteorological conditions and depictions of each farm layout can be found in Arndt et al. (2018).

### 3.2   Tracer flux emission estimates via aircraft

Some plumes appear to represent the entire site and all of its sub-sources ("whole-site"). Other plumes occur when the wind direction isolates an individual source (e.g. animal housing) for a given transect. For these close and fast transects, it can be difficult to have the tracer in a position that appears to adequately represent the site or a sub-source. Designating each observed plume to a source considers many factors but is ultimately up to the discretion of the analyst. Efforts to understand this interpretive bias are described in the Supplementary Information, and use two validation methods, one analyst-driven and one automated.

Whole-site plumes averaged for each farm closely resemble estimates by other methods during the initial Arndt et al. (2018) study (Table 2), and fall within the uncertainty of all methods for both farms. Emissions associated with animal housing (based on tracer proximity and wind direction) resemble mobile laboratory findings. Animal housing



emission rates cannot be directly compared to the results of the mass balance technique from the original study as there was no apportionment by source (only whole-site estimates).

### 3.3 Overlapping measurements between platforms

Occasionally, the aircraft flew over the mobile lab while both vehicles were recording data. One example of this coincidence can be seen in Fig. 3, allowing for direct comparison between these two methods. Around midday of June 22 2016, the aircraft (11:41:25 – 11:41:50 PDT) and the minAML (11:40:45 – 11:42:00 PDT) encountered the tracer gas and site emission plumes for 25 seconds and 75 seconds respectively. For this section of flight, the aircraft flew at around 74 m s$^{-1}$ (165 mph) covering 1.3 km (0.8 mi). Meanwhile, the minAML drove on a paved road at about 16 m s$^{-1}$ (35 mph) over 0.8 km (0.5 mi). Both transects occurred in the same direction, from east to west to the south of the site. During the overlapping transects, each platform saw a sharp increase in $CH_4$ concentration followed by a broad enhancement at lower concentrations while a similarly rapid rise in $C_2H_6$ concentration was followed by steady decrease. Differences in baseline values of $CH_4$ and $C_2H_6$ are attributed to different schedules of acquiring backgrounds (inlet overblown with zero air more frequently on the minAML). Given the similar spatial characteristics of these plumes, it seems likely both platforms were observing the same airmass. As expected, the aircraft-based observations show a lower temporal resolution versus the mobile lab due to speed differences. While these plumes would not be used for emission estimations based on tracer ratio due to poor correlation, they show that the same airmass can be sampled on the ground and in the air.

## 4 Discussion

### 4.1 On-site sampling by aircraft

Tracer flux ratio methodology thrives with strong winds and downwind road access perpendicular to the dominant wind direction. Close placement of tracer gas to a point source and distant measurements by the mobile lab allow time and space for the tracer to co-disperse with emission gas and merge together in the measured plume. During this field campaign, the aircraft flew close to the site measuring emissions in a calm wind and saw an abundance of signal due to strong surface heating. These conditions proved favorable for the aircraft and mass balance calculations but stretch the possible application of the tracer release method. Even so, the attempt to perform a tracer release experiment observed from an aircraft proved largely successful and provided direct insight as to how these measurements relate to the ground-based observations.

Due to the sensitivity of the $C_2H_6$ instrument on the aircraft, it was readily apparent when the tracer gas was present and intermingling with the farm emissions. As seen in Fig. 4, prior to releasing any tracer gas, the concentration of $C_2H_6$ shows a relatively steady baseline. After initiating the release of tracer gas at 20 slpm, it took approximately 20 minutes before the aircraft begins to detect it. The aircraft ascended above the emission plume for 10-20 minutes after the release began, which may have lengthened the time it took to first detect tracer gas. After being observed on-board the aircraft, another 15 minutes pass before transects reliably encounter emissions with corresponding tracer gas, which we attribute to a combination of decreasing altitude and evolving dispersion.





Emission magnitudes showed no bias with altitude change at Dairy 2, though a slight negative correlation existed at Dairy 1 ($R^2 = 0.25$). Increasing emissions with decreasing height, in some cases, could be attributed to the influence of a strongly lofted lagoon signal at this site. Lower flights could have caused the aircraft to encounter a larger proportion of the manure-related emissions instead of the ideal case: a well-mixed plume representative of the entire

site. Animal housing emission estimates saw no significant correlation with altitude at Dairy 1.

During each flight, identifiable plumes of $CH_4$ were observed at a steady pace (approximately every 1-2 min). As seen in Fig. 5, repeated plumes also revealed unique characteristics within a site. Viewed from the south, manure and animal housing areas at Dairy 1 line up together, whereas at Dairy 2, the anaerobic lagoon and settling cells are offset from the housing areas. While these observations largely depend on wind direction and distance from the source, some

features gave insight into where emissions came from on-site. Broad emissions can be readily attributed to the large collection of point source emitters milling around barns and open lots (cows of various ages). Sharp peaks and broad plateaus indicate an encounter with outgassing by a large area source (liquid manure ponds). Gaussian shapes appear to be an amalgamation of both sub-sources mixed downwind.

Temporal and spatial differences exist between the aircraft measurements used in this dataset and the ground-

based measurements collected as part of the initial study (Arndt et al., 2018). Measurements by the minAML occured during the day and night at a variety of distances from each site (up to 6 km). The aircraft had good coverage during the middle of the day, with flights in the late morning and early afternoon performing frequently repeated transects above each site (~1 km). The ground-based tracer release experiment struggled at times in the hot mid-day conditions due to low winds and strong vertical mixing while the aircraft saw good signal, but had no issue collecting night-time

measurements.

### 4.2  Experimental challenges

Swirling and calm winds shifted emissions around each site at various times over multiple days. When selecting valid plumes, proximity of the aircraft during an enhancement to a single sub-source introduces a dilemma. Varying distances between the tracer gas release point and presumed source could affect the determined emission rate, due to

imperfect co-dispersion. For example, using a tracer plume located 500 m away to represent a source 300 m away would be problematic. When measuring at greater distances with better resolution (due to sampling in a slower vehicle), it is often trivial to identify when the tracer inadequately represents the emission. Flying several times faster than the driven transect provided notable repeatability but made spatial understanding of the site difficult with respect to emission sources.

Direct estimates of liquid manure emissions proved unrealistic at both dairies due to sparse amounts of $CH_4$ plumes with sufficient tracer representation, despite favorable wind direction and aircraft position. A few plumes of acceptable data quality were identified as being related to liquid manure emissions at Dairy 2 (n = 4), but estimates were significantly higher than reported in Arndt et al. (2018) at $5{,}166 \pm 837$ kg $CH_4$ d$^{-1}$. Due to concerns that the tracer release location was not close enough to the liquid manure source to be representative, especially due to nonideal

transect geometry and limited horizontal wind, this data is not reported in Table 1. Relative apportionment of $CH_4$ between sub-sources (using only whole-site and animal housing values) showed manure-associated plumes leading





the fractional contribution at Dairy 1 (69:31) and Dairy 2 (55:45). This was an expected finding based on US EPA methodology estimates (Arndt et al., 2018) for this month at Dairy 2 (73:27). Given the temporal natural of manure emissions, as reported by Leytem et al. (2017), it should be reinforced that these results only represent a short period of time (6 measurement days) in a single season. Despite the difficulty of collecting or identifying many distinct

manure-associated plumes via measurements taken from this aircraft, the general apportionment of source emissions appears to remain evident.

     Clear hot measurement days could have stimulated anaerobic activity in manure lagoons and caused greater release of gases (Safley et al., 1988), while strong thermal convection lofted concentrated and unmixed plumes. Aside from refinements to the method (e.g. moving the tracer gas closer to the source), performing this technique in different

seasons, meteorological conditions, and during mixing events (e.g. flushing) would enhance our understanding of the variability in emissions from liquid manure management on dairy farms.

     For the mobile laboratory, road access was a challenge at times. Large plots of surrounding cropland typically had a limited number of roads crossing through them, with those available often being private or undeveloped. In order to collect plumes adequately downwind of each site on accessible public roads, the ground-based ARI team

required winds to come from certain directions. Being able to fly above the site eliminates these challenges. However, the aircraft flew a set pattern at each site, circling a particular radius to optimize the established mass balance method, and did not explore downwind like the vehicle. As seen in Fig. 6, plumes used for determining emission rates were clustered in areas above each site that typically agreed with the dominant wind directions along the looping flight path. Wind rose plots for each site represent the wind conditions observed by the aircraft during the midpoint of each plume

event (Fig. 6C and 6D). On-site wind measurements during these events provided additional insight as to how the wind evolved between the site and aircraft. Other plume events sometimes occurred inside of the dominant downwind fetch, especially during calm wind conditions, but lacked the prerequisites to be included in emission estimations.

### 4.3 Future work

Future work towards refining the tracer release method to an aircraft will require several improvements to the current

experimental design. Instead of flying around the perimeter of a dairy farm or other emission source in a circle as part of an established mass balance approach (Conley et al., 2017), the aircraft should mimic the driven transects of the mobile lab via long horizontal transects at varying distances perpendicular to the dominant wind direction (Hacker et al., 2016). Conducting downwind transects at greater distances (e.g. 500 m to 5 km) would allow for better comparisons between platforms but may not be feasible in conditions similar to those experienced in this study (strong

surface heating combined with calm horizontal winds), as it could be difficult to encounter the plume.

     Rather than relying on only a couple point source releases, tracer gas could be released as a line or grid source along the border of liquid manure management areas or animal housing fence lines (Lamb et al., 1995; McGinn et al., 2006). Increasing the flow rate of tracer rate from 15 slpm by several factors would improve signal-to-noise ratios of tracer enhancements. Furthermore, an aircraft carrying a second instrument on-board that quickly (1 Hz) and precisely

(ppt sensitivity) monitors a second tracer gas (e.g. $C_2H_2$) would provide a check on the observed tracer concentrations or could aid source identification. With two tracer gases, the initial ratio of release rates ought to persist throughout





the migration of the plumes and be reflected in the ratio of downwind enhancements ("dual tracer ratio"; Roscioli et al., 2015). Deviations from the expected value indicate loss of tracer gas and inadequate representation of a source. It should be noted that the two tracers used in this original study were employed as independent tracers for better coverage over large multisource areas, while the scenario described above applies to overlapping use of tracer gases

(two tracers for a single source). Benefits of adding a second tracer (dual-tracer flux ratio methodology) are described further in Roscioli et al. (2015).

Overall, combining these measurement techniques through aircraft-observed tracer release promotes positive aspects of each method. Low-flying aircraft measurements occur rapidly on a versatile platform with no road access restrictions. Tracer gases can indicate sources, identify interferences, and enable quantification without relying on

modelling or highly accurate wind measurements. Using this method, an aircraft can have greater confidence identifying sources and can confirm ground-based observations.

## 5   Conclusion

By quantifying $CH_4$ emissions to within the uncertainties of stand-alone ground-based tracer and aircraft mass balance measurements, this study demonstrates the viability of performing a tracer release experiment from the ground

observed by an aircraft flying overhead. Other than intentionally overlapping measurement times, we were able to demonstrate a third method of monitoring dairy emissions using data collected for previously established techniques, without prior coordination or making any procedural changes in the field. In this case, an aircraft flying transects prioritized for a mass balance methodology successfully collected data viable for single tracer flux ratio analysis. Simultaneous observations by the aircraft and mobile laboratory on a similar spatial scale provide a brief look into of

how each technique experiences single tracer flux ratio methodology. Considering the success in applying this method, a refined approach could greatly improve and further demonstrate the feasibility of this technique.








*Data availability.* A table containing Site IDs, source designations, measurement durations, $CH_4$ emission rates, plane altitude, measurement distance from the point of tracer release, wind direction and speed, and coefficients of determination for $CH_4/C_2H_6$ can be found in Supplementary Materials (SM Dataset) as a Microsoft Excel file.

*Author contribution.* CA coordinated the field campaign. SC and ICF collected the aircraft data. CD, TIY, and JRR participated in the mobile laboratory measurements. CD, TIY, JRR, SC, ICF, and SCH performed data interpretation and analysis. CD prepared the manuscript with contributions from ICF. All authors critically reviewed the manuscript. All authors approved the submitted version for publication.

*Competing interests.* The authors declare that they have no conflict of interest.

*Acknowledgements.* C. Arndt's postdoctoral fellowship at Environmental Defense Fund was funded by a gift from Sue and Steve Mandel and the Kravis Scientific Research Fund (New York, NY). The measurements were funded by a gift from Sue and Steve Mandel and the Robertson Foundation (New York, NY). I. Faloona's effort was supported by

the USDA National Institute of Food and Agriculture, [Hatch project CA-D-LAW-2229-H, "Improving Our Understanding of California's Background Air Quality and Near-Surface Meteorology"].

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



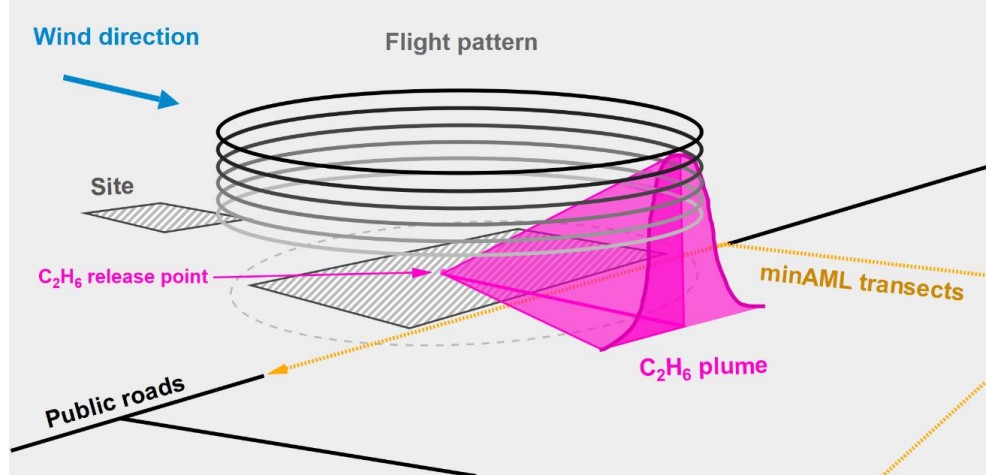

**Figure 1. Experimental schematic of tracer release (ethane; $C_2H_6$) at a dairy farm, as observed by a small aircraft and miniature Aerodyne Mobile Laboratory (minAML). In this ideal scenario, the wind is carrying across the site perpendicular to accessible public roads.**



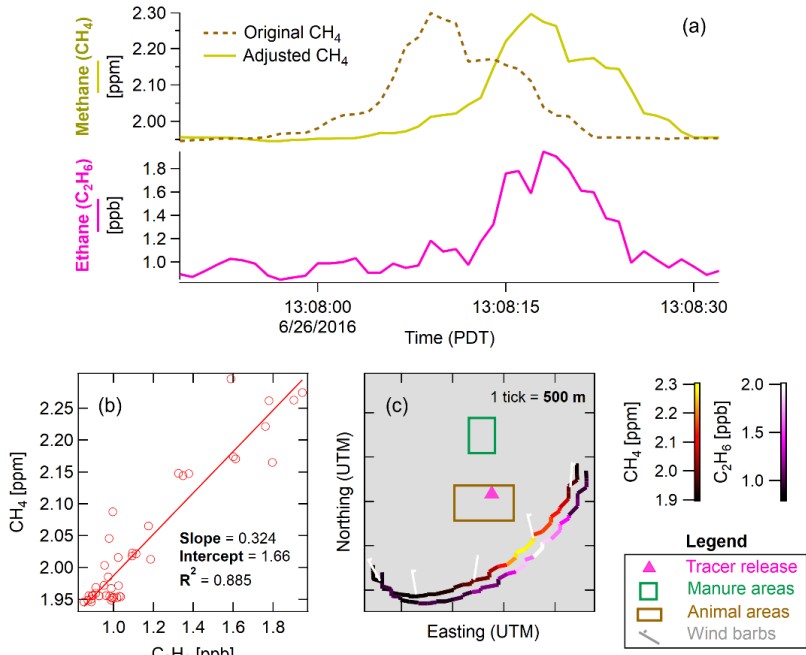

**Figure 2. Time traces of methane (CH₄) and ethane (C₂H₆) during a flight around Dairy 1 (panel a). A correlation plot with a best-fit line (panel b) compares enhancements above baseline of CH₄ and C₂H₆ (panel a) after accounting for differences in instrument response times and tracer position relative to site emissions. A map of Dairy 1 overlaid with the flight path is colored by CH₄ concentration (panel c). An identical transect colored by C₂H₆ is offset slightly for clarity. Wind barbs depict the wind velocity (averaging 5 knots from NNW) at several points during the transect.**


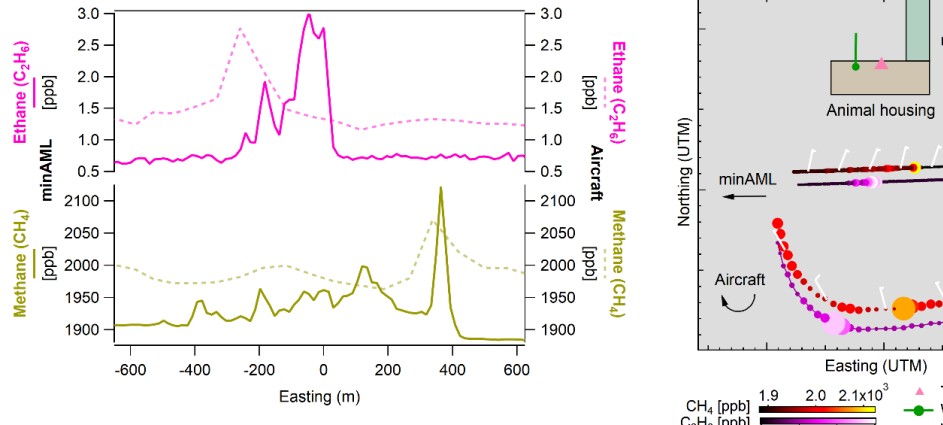

**Figure 3. Plumes observed by the miniature Aerodyne Mobile Laboratory (minAML) and aircraft. Plots of methane (CH₄) and ethane (C₂H₆) are overlaid for each platform (panel a). Observations occurred during transects by each vehicle to the south of Dairy 2, during a release of C₂H₆ into a southerly wind (panel b). Potential emission sources on the farm have been identified as colored sections, though not as an exact scaled representation.**





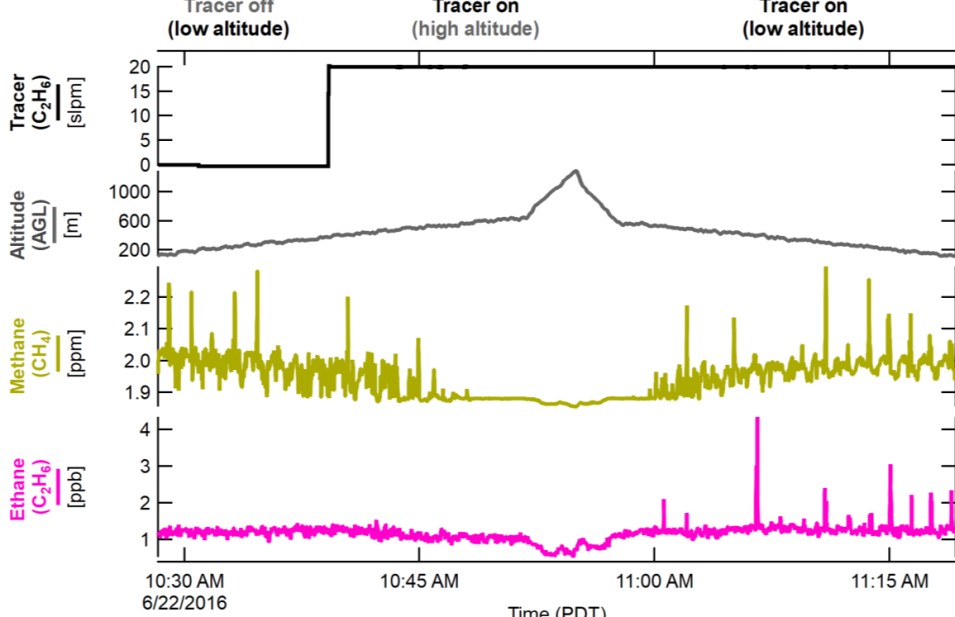

**Figure 4. Comparison of flight sampling periods prior to and during release of tracer gas (ethane, $C_2H_6$), showing enhancements of methane above Dairy 2 with and without corresponding peaks of $C_2H_6$ depending on release rate, altitude (AGL), and dispersion.**





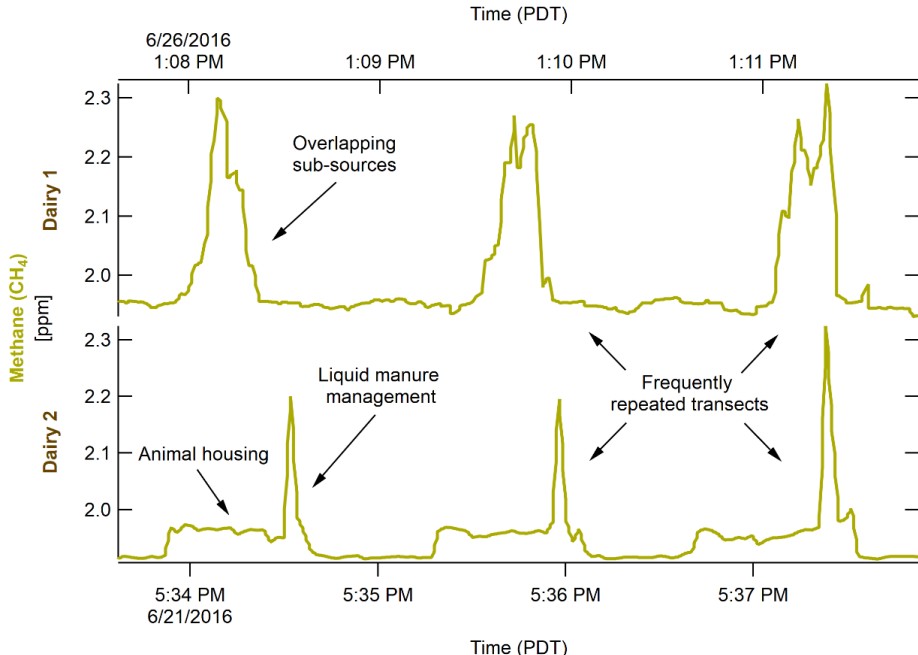

**Figure 5.** Selected sampling periods (approximately 5 minutes) at each dairy farm showing characteristics of emitted methane plumes as observed by the aircraft downwind to the south. Each time trace depicts the high rate of repetition in the flown transects around each site.





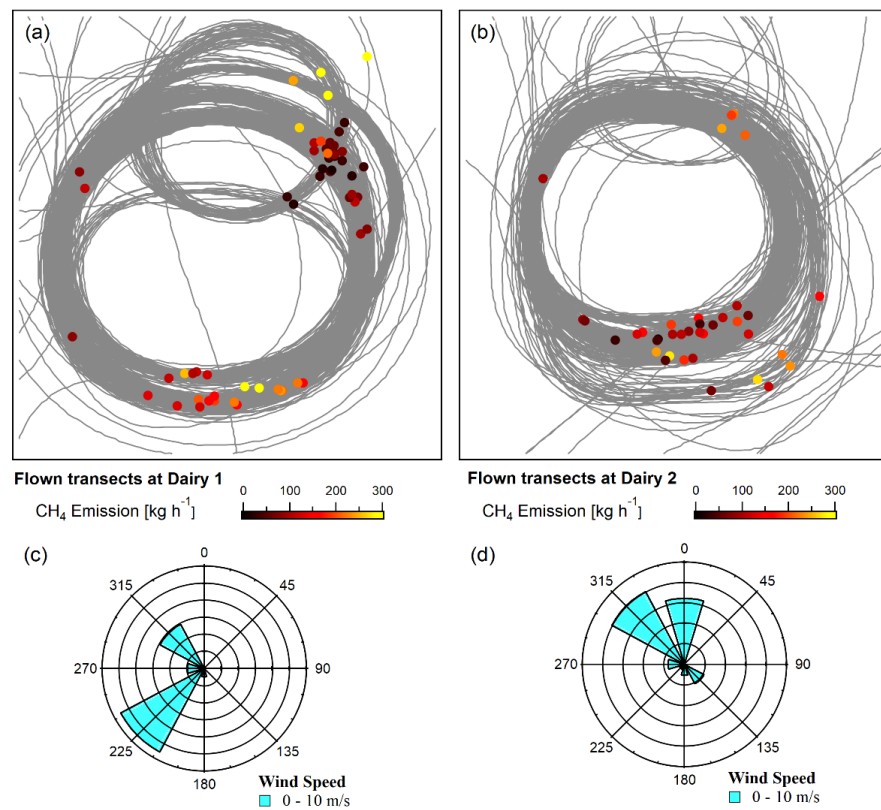

**Figure 6. Methane (CH₄) emission rates displayed on every flight track as dots, positioned at the mid-point of each enhancement event (panel a and b). Corresponding wind roses average the originating direction and magnitude of the wind from the mid-point of each plume event (panel c and d).**



**Table 1. Overlap between flight times and release of tracer gas (ethane) over the course of the field campaign.**

|  | Days spent [n] | Total release | Overlap | Overlap by flight |
|---|---|---|---|---|
|  |  | [Elapsed time - hh:mm] | | |
| Dairy 1 | 5 | 13:00 | 3:55 | 0:47 |
| Dairy 2 | 6 | 27:05 | 7:25 | 1:14 |
| Both Sites | 11 | 40:05 | 11:20 | - |

*Release on June 25th but no flights.

25

30





**Table 2. Comparison of methane emission estimates (kg d$^{-1}$ ± 95% C.I.) for two dairy farms between this paper ("tracer-plane") and established tracer release ("ARI") and mass balance ("SA") methods.**

| Source | Tracer-Plane | ARI[*] | SA[*] |
|---|---|---|---|
| | [kg CH$_4$ d$^{-1}$ ± 95% C.I.] | | |
| Whole-site | | | |
| Dairy 1 | 5,850 ± 793 | 6,985 ± 626 | 7,249 ± 2,153 |
| Dairy 2 | 3,699 ± 685 | 3,046 ± 814 | 3,274 ± 745 |
| Animal housing | | | |
| Dairy 1 | 1,835 ± 295 | 2,601 ± 811 | - |
| Dairy 2 | 1,648 ± 563 | 1,636 ± 513 | - |
| Liquid manure | | | |
| Dairy 1 | - | 5,994 ± 579 | - |
| Dairy 2 | - | 2,141 ± 637** | - |

*Arndt et al., 2018.

**Settling basin value only, from Arndt et al., 2018.