# Peer review of "Using the tracer flux ratio method with flight measurements to estimate dairy farm CH4 emissions in central California"

_Atmospheric Measurement Techniques, 2018_

## Referee Comment (RC1) · Anonymous Referee #2 · 5 Dec 2018

In this paper, the author mix up a well-known technique of estimating emission, the tracer release method with airplane measurement usually used for mass-balance approach. They show that using an aircraft to do the tracer method is feasible even when the original campaign was not exactly designed for it and that the validity and precision of the results could be improved if the airplane flight paths are adapter to the tracer ratio method. This is an interesting and well-written paper and we recommend publication after minor changes.

P1 l19-21: The last comment is not really necessary or should be reformulated.

p3 Instruments for the ground-based transects are described but the one that data

are used from (except the TILDAS) are not detailed. please, add some details about instrumentation for CO2, wind, ... for the airplane as well.

p6 l2 It would be good here already to say why there is no estimate for the manure as we are expecting it from the previous sentences.

Table 1: there is no * that relates to the comment below the table. Either erase it or modify the table accordingly

Figure 2: Please use SI units and not knots

[Figure]

---

## Referee Comment (RC2) · Anonymous Referee #1 · 7 Dec 2018

general comments

Daube et al. compare three methods of estimating methane emissions from a dairy farm. Two are well-established in the scientific literature: tracer release using a mobile laboratory and mass balance using an aircraft. The third is tracer release using an aircraft. They find generally good agreement between emissions determined using the tracer release/aircraft combination and the other two methods.

This paper extends the work of tracer release using mobile laboratories into the realm of airborne measurements. As such, it is a relatively straightforward analysis comparing the two methods. The only thing I find lacking is a more in-depth discussion of the

added complexity of the additional dimension, the vertical, that the aircraft can probe compared to the mobile laboratory. I think a plot of altitude vs. the CH4/ethane ratio might be informative. The authors briefly described this in the first paragraph of page 7, but I think a visual presentation might be better.

I also question the usage of a linear regression because of the need to align the two plumes. The tracer is presumably a point source, but a manure lagoon might be meters or tens of meters wide. I would expect sources such as this to have much broader plumes than the tracer release. Or worse yet, there could be two separate methane plumes, but only one ethane plume. The only way to properly account for this is by taking the integral. For example, what is the ratio of the integrals for the transect shown in Figure 2?

In Table 2, why do the emissions from Animal Housing and Liquid Manure add up to more than the Whole-site emission?

specific comments

The authors should be consistent with their use of the term "sub-source". First, they should probably define it, because I'm not sure it's a word. Second, I would prefer they simply use the word "source", and remind the readers that a dairy farm is actually a combination of many different sources: cows, manure, settling cells, etc.

For Figures 2 and 3, I would clearly label (or title) these as Dairy 1 and Dairy 2, instead of burying this information in the caption.

p. 1, line 25: is that 60% of anthropogenic CH4 emissions?

p. 2, line 27: how "low and close" did the aircraft fly?

p. 6, line 8: what altitude was the aircraft at for the transect described here?

p. 6, line 28: I think this section needs a little introduction, instead of immediately delving into the ethane background. Which dairy is this? Introduce whatever Figure 4

is before referring us to it.

Figure 4 and p. 6, line 31: Based on horizontal wind speed, how long would you expect it to take to see the signal from the aircraft after the tracer had been released?

technical corrections

p. 2, line 12: should that reference be for "Grainger"?

p. 3, line 1: since ethane isn't emitted from dairy farms, I suggest saying "... emitted from within dairy ..."

p. 4, line 15: add "its" to read "... and its long atmospheric lifetime."

p. 4, line 21: change to "molar enhancement ratio" and add comma after "The molar ratio," p. 5, line 16: replace "generally" with "with speeds"

p. 5, line 26: why say "appear to represent"? I would assert that they do represent an entire site.

p. 5, line 32: "plumes" isn't really the noun you should use in this sentence. You are talking about emission estimates here.

p. 6, line 4: instead of "recording data", perhaps say "sampling the same plume"

p. 6, line 6: add comma after "22"

p. 7, line 30: I suggest "sparse number" instead of "sparse amounts"

p. 8, line 16: I suggest "circling at a particular radius"

p. 9, line 13: I suggest "independent" instead of "stand-alone"

p. 9, line 19: remove "of"

p. 13, line 3: I suggest "the wind is carrying the plume across the site..."

---

## Author Comment (AC1) · 7 Dec 2018

**Response to Referee #2**

We would like to thank the referee for their helpful comments and kind words. We have responded below with referee comments in red and our response in black. We have also included a marked-up version of the revised manuscript to this response.

In this paper, the author mix up a well-known technique of estimating emission, the tracer release method with airplane measurement usually used for mass-balance approach. They show that using an aircraft to do the tracer method is feasible even when the original campaign was not exactly designed for it and that the validity and precision of the results could be improved if the airplane flight paths are adapter to the tracer ratio method. This is an interesting and well-written paper and we recommend publication after minor changes.

P1 l19-21: The last comment is not really necessary or should be reformulated.

We agree, and have removed the last sentence of the abstract.

p3 Instruments for the ground-based transects are described but the one that data are used from (except the TILDAS) are not detailed. please, add some details about instrumentation for CO2, wind, ... for the airplane as well.

We have added detail about instruments used on the aircraft during this campaign: "Scientific Aviation equipped an aircraft with a Picarro (Santa Clara, CA) G2301-f cavity ring-down spectrometer ($CO_2$, $CH_4$, $H_2O$), Vaisala (Helsinki, Finland) HMP60 Humidity and Temperature probe, and Hemisphere VS330 GPS Compass used for positioning and calculating wind velocity (Conley et al., 2014).".

p6 l2 It would be good here already to say why there is no estimate for the manure as we are expecting it from the previous sentences.

We agree, and have added a sentence to address the lack of a manure estimate: "Measurements of manure emissions were not compared with established techniques due to uncertainty in representation of the source by the tracer gas.".

Table 1: there is no * that relates to the comment below the table. Either erase it or modify the table accordingly

We have associated the * to the "Days Spent" column in Table 1 to indicate that one of the days spent on-site releasing tracer gas was not accompanied by a measurement flight.

Figure 2: Please use SI units and not knots

We have changed the unit in the caption of Figure 2 to be in "m s$^{-1}$" and converted the value accordingly.

[revised manuscript text omitted]

---

## Author Comment (AC2) · 4 Feb 2019

**Response to Referee #1**

We thank the referee for their insightful comments and questions. We have responded below with referee comments in red and our response in black. We have also included a marked-up version of the revised manuscript to this response.

**General comments**

10 Daube et al. compare three methods of estimating methane emissions from a dairy farm. Two are well-established in the scientific literature: tracer release using a mobile laboratory and mass balance using an aircraft. The third is tracer release using an aircraft. They find generally good agreement between emissions determined using the tracer release/aircraft combination and the other two methods.

15

5

This paper extends the work of tracer release using mobile laboratories into the realm of airborne measurements. As such, it is a relatively straightforward analysis comparing the two methods. The only thing I find lacking is a more in-depth discussion of the added complexity of the additional dimension, the vertical, that the aircraft can probe compared to the mobile laboratory. I think a plot of altitude vs. the CH4/ethane ratio might be informative. The authors briefly described this in the

20 of altitude vs. the CH4/ethane ratio might be informative. The authors brief first paragraph of page 7, but I think a visual presentation might be better.

We agree that a plot of emissions by altitude would enhance discussion of the subject. The plot below uses all the plumes from each dairy to relate aircraft altitude to CH4 emission rates.
Emissions between 0 – 6,500 kg day-1 appear to be randomly distributed between 100 – 600 m at each site. A couple outliers show higher emission rates at low altitudes and a few others are scattered across a range of emission rates for higher altitude observations. We have added this figure (Fig. 6) and relevant discussion to Section 4.1.

I also question the usage of a linear regression because of the need to align the two plumes. The tracer is presumably a point source, but a manure lagoon might be meters or tens of meters wide. I would expect sources such as this to have much broader plumes than the tracer release. Or

35 worse yet, there could be two separate methane plumes, but only one ethane plume. The only way to properly account for this is by taking the integral. For example, what is the ratio of the integrals for the transect shown in Figure 2?

The reviewer is correct that we failed to discuss this important alternative analysis. We have addressed this concern by performing a reanalysis of the dataset using integration of plumes to determine emission rates. We acknowledge the mismatch of point source release being used for area source representation and recognize the need to seek different release methods for future similar studies of area sources – line source, area source via a grid arrangement, etc. We had employed multiple tracers and relied on greater sampling distances in the original study to obtain better tracer representation as noted in the paper, but obviously those factors don't contribute to

this analysis.

In general, we look at discrepancies in line shape as a first indicator of poor tracer representation of a source. In Figure 2, the plumes of  $CH_4$  and  $C_2H_6$  have similar structure with no unaccounted for  $CH_4$  peaks nearby, signifying adequate co-dispersion and tracer representation for this event.

for CH4 peaks nearby, signifying adequate co-dispersion and tracer representation for this event. When we perform linear regression, we match offset plumes in time (within a reasonable time range) based on multiple linear regression tests before determining the optimal timing and resulting correlation. The ratio of the integrated areas for the enhancements shown in Figure 2 results in an emission rate of 223 kg hr-1 for a tracer release rate of 15 slpm. Using the slope of the linear regression, this emission rate is calculated to be 203 kg hr-1 at the same tracer release rate.

During the original analysis, we had concerns regarding the potential bias in selecting a baseline, especially in cases of low signal-to-noise. For example, in determining a baseline for the  $C_2H_6$ plume from Figure 2, altering one side of the baseline from the ideal case by 50 ppt resulted in a 20 kg hr-1 change in emission rate. Using slopes from linear regression for these plumes provided a systematic unbiased approach for relating these enhancements of CH4 and tracer gas. However, linear relationships can be misleading if taken at face value (i.e. high R2 value) and modelling of data via linear slopes need to be considered in the context of the entire enhancement event.

Analysing the entire dataset again using integration compared quite favourably to our original analysis. This serves as a valuable check on the use of linear regression for this application, as discrepancies between the two methods could imply weak co-dispersion or misleading correlations.

We have replaced values in Table 2 with those determined by integration in the reanalysis. Three plumes were removed from Dairy 2 and two were removed from Dairy 1 observations keeping in accordance with quality control described in Section 2.3.

In Table 2, why do the emissions from Animal Housing and Liquid Manure add up to more than the Whole-site emission?

75

Emission estimates of Dairy 1 using the tracer release method via a ground-based mobile laboratory are presented in Table 2 for the overall site and two major sources from Arndt, et al., 2018. Whole-site emission rates fall within the uncertainty of the summed source emission rates. In ranges based on the 95% confidence intervals, the whole-site emission rates range 6,349 to

80 7,611 kg CH4 hr-1. Animal housing emission rates range 1,790 kg to 3,412 kg CH4 hr-1. Liquid manure emission rates range 5,415 to 6,573 kg CH4 hr-1.

**Specific comments**

- 85 The authors should be consistent with their use of the term "sub-source". First, they should probably define it, because I'm not sure it's a word. Second, I would prefer they simply use the word "source", and remind the readers that a dairy farm is actually a combination of many different sources: cows, manure, settling cells, etc.
- 90 We have replaced all mentions of the term "sub-source" with "source", as suggested.

For Figures 2 and 3, I would clearly label (or title) these as Dairy 1 and Dairy 2, instead of burying this information in the caption.

95 Corrected.

p. 1, line 25: is that 60% of anthropogenic CH4 emissions?

Yes, we have added "anthropogenic" to this sentence (Figure 4, page 56 of CARB, et al. 2017).

**p. 2, line 27: how "low and close" did the aircraft fly?**

We have added average distance altitude values from flights at both sites. Section 3.1 contains details about maxima and minima altitudes.

105

115

100

p. 6, line 8: what altitude was the aircraft at for the transect described here?

The average altitude of the aircraft was 428 m AGL during this 34 s plume.

110 p. 6, line 28: I think this section needs a little introduction, instead of immediately delving into the ethane background. Which dairy is this? Introduce whatever Figure 4 is before referring us to it.

We have introduced Figure 4 and noted which dairy farm is being shown in the paragraph as well as in Figure 4 (Dairy 2). We have also changed the order of the paragraphs in Section 4.1 to help the flow of this section and add clarity.

Figure 4 and p. 6, line 31: Based on horizontal wind speed, how long would you expect it to take to see the signal from the aircraft after the tracer had been released?

Based on the average wind direction (from the NW) and horizontal speed (4.2 m s-1) from 10:39 AM PDT (start of tracer release) to 11:00 AM PDT (first spike of  $C_2H_6$ ), we would expect to begin

seeing tracer gas after  $\sim 6$  minutes at a distance of 1.6 km (from release point to the intersection between the circular transect and wind direction). We saw the first spike shortly after when would have expected, around 11 minutes after beginning release.

125

**Technical corrections**

p. 2, line 12: should that reference be for "Grainger"?

130 Yes, thank you for catching that typo.

p. 3, line 1: since ethane isn't emitted from dairy farms, I suggest saying ". . . emitted from within dairy . . ."

135 Agreed, we have added the word "within" to the sentence.

p. 4, line 15: add "its" to read ". . . and its long atmospheric lifetime."

Corrected.

140

```
p. 4, line 21: change to "molar enhancement ratio" and add comma after "The molar ratio,"
```

Corrected.

145 p. 5, line 16: replace "generally" with "with speeds"

Corrected.

- p. 5, line 26: why say "appear to represent"? I would assert that they do represent an entire site.
- 150

Agreed, we have changed language in this paragraph to reflect this comment.

p. 5, line 32: "plumes" isn't really the noun you should use in this sentence. You are talking about emission estimates here.

155

165

Agreed, we have changed "plumes" to "emission estimates".

p. 6, line 4: instead of "recording data", perhaps say "sampling the same plume"

**160 Corrected.**

p. 6, line 6: add comma after "22"

Corrected.

p. 7, line 30: I suggest "sparse number" instead of "sparse amounts"

Corrected.

p. 8, line 16: I suggest "circling at a particular radius"

Corrected.

175

Corrected

p. 9, line 19: remove "of"

180 Corrected.

p. 13, line 3: I suggest "the wind is carrying the plume across the site. . ."

Corrected.

185

190

195

200

205

**Using the tracer flux ratio method with flight measurements to estimate dairy farm CH4 emissions in central California**

215 Conner Daube1, Stephen Conley2,3, Ian C. Faloona3, Claudia Arndt4, Tara I. Yacovitch1, Joseph R. Roscioli1, Scott C. Herndon1

1Aerodyne Research, Inc., Billerica, MA, 01821, USA
2Scientific Aviation, Inc., Boulder, CO, 80301, USA
3Department of Land, Air, and Water Resources, University of California, Davis, CA, 95616, USA
4Environmental Defense Fund, San Francisco, CA, 94105, USA

220

*Correspondence to*: Conner Daube (daube@aerodyne.com)

Abstract. Tracer flux ratio methodology was applied to airborne measurements to quantify methane (CH4) emissions from two dairy farms in central California during the summer. An aircraft flew around the perimeter of each farm measuring downwind enhancements of CH4 and a tracer species released from the ground at a known rate. Estimates of CH4 emission rates from this analysis were determined for whole sites and major sub-sources within a site (animal housing and, liquid manure lagoons). Whole-site CH4 flux rates for each farm, Dairy 1 (6,1085,850 ± 821793 kg CH4 day-1, 95% confidence interval) and Dairy 2 (4,0183,699 ± 456685 kg CH4 day-1, 95% confidence interval), closely resembled findings by established methods: ground-based tracer flux ratio and mass balance. Individual Sub-ssource emission rates indicate a greater fraction of the whole-site emissions come from liquid manure management than animal housing activity, similar to bottom-up estimates. Despite differences in altitude, we observed that the tracer release method gave consistent results when using ground or air platforms.

230

225

**1** Introduction**

[revised manuscript text omitted]

---

## Author Response (AR2)

**Response to the editor**

We thank you, Dr. Huilin Chen, for your efforts as the editor of this manuscript and appreciate your feedback. We have presented your comments below in red, relevant sections of the manuscript in blue, and added our comments and proposed changes in black text.

**Publish subject to minor revisions (review by editor)**

Comments to the Author:
Dear authors,

Given that you have significantly improved the manuscript according to both reviewers' comments, I gladly accept your manuscript for publication after addressing the suggestions for revision from Referee #1. Besides these, the two outliers in figure 6 (Dairy 1) at low altitudes were attributed to not well-mixed conditions. What were the wind conditions associated with those two release tests? Does the shape of the plumes provide any support for your claim?

Best regards,
Huilin Chen

**Original text from manuscript:**

For the plumes reported in this dataset, there is no observed dependence of emission rate with sampling altitude. In Fig. 6, $CH_4$ emissions are plotted versus aircraft altitude. Emissions between $0 – 6,500$ kg d$^{-1}$ appear to be randomly distributed between $100 – 600$ m at each site (Fig. 5). Two outliers show higher emission rates at low altitudes, unmatched at higher altitudes. Above 650 m are three other points scattered across a wide range of emissions ($2,000 – 6,500$ kg). Increasing emissions with decreasing height, in some cases, could be attributed to the influence of a strongly lofted lagoon signal at a site. Lower flights could then cause the aircraft to encounter a larger proportion of the manure-related emissions instead of the ideal case: a well-mixed plume representative of the entire site.

**Author comments:**

Wind conditions during these plumes were from the SW and SSW (217 and 205 deg.) with horizontal wind speeds at 5.5 and 5.9 m s$^{-1}$ respectively. Tracer plume shapes indicate imperfect co-location and differences in dispersion with respect to broadening and temporal offsetting when compared to their corresponding emission plume shapes. That being said, both resemble each other characteristically enough to have been considered for assignment. In both cases, the aircraft flew closer to the source than the tracer and this case of disparate dispersion conditions can affect emission estimates. We propose a concise but necessary addition of two sentences further detailing the outlier conditions to the original paragraph in the section below (see "Proposed changes"). We cite a 2015 paper by Goetz, et al. which goes into depth in the text and Supplementary Information describing the need to account for tracer-source distances when performing the tracer flux ratio method.

**Proposed changes:**

Pg 7, ln 35: Addition of the following line:

"These outliers occurred when the aircraft flew close to the site at an angle that put the lagoon between the aircraft and the tracer release point. The impact of measuring a source closer than the tracer is a potential overestimation of the emission due to differences in dispersion (Goetz et. al., 2015)."

Pg. 12, ln 15: Added citation for Goetz et al., 2015.

"Goetz, J. D., Floerchinger, C., Fortner, E. C., Wormhoudt, J., Massoli, P., Knighton, W. B., Herndon, S. C., Kolb, C. E., Knipping, E., Shaw, S. L., and DeCarlo, P. F.: Atmospheric Emission Characterization of Marcellus Shale Natural Gas Development Sites, Environ. Sci. Technol., 49, 7012-7020, https://doi.org/10.1021/acs.est.5b00452, 2015."

**Minor edits:**

Pg 2, ln 22: Removed a space from the typo: "measurements  in".

Pg 3, ln 19: Removed a space from the typo: "SA  had".

Pg 3, ln 36: Removed a space from the typo: "relevant .".

Pg 5, ln 6: Added a space to the typo: "site.At".

Pg 5, ln 24: Removed a space from the typo: "For  close".

Pg. 12, Ln 36: Fixed broken doi link to "https://doi.org/10.1021/es506352j".

**Marked-up manuscript:**

[revised manuscript text omitted]